# Violence, city size and geographical isolation in African cities

**Rafael Prieto-Curiel** [1] **& Ronaldo Menezes** [2,3] ✉

Different types of violence are commonly linked with large urban areas, often presumed to scale superlinearly with population size (i.e., to be disproportionately higher in larger cities). This study explores the hypothesis that smaller, isolated cities in Africa may experience a heightened intensity of violence against civilians. It aims to investigate the correlation between the risk of experiencing violence, a city's size and its geographical isolation. Between 2000 and 2023, incidents of civilian casualties were analysed to assess lethality in relation to varying levels of isolation and city size. African cities are categorised by isolation (measured by the number of highway connections) and centrality (the estimated frequency of journeys). We show that violence against civilians exhibits a sublinear pattern, with larger cities witnessing fewer events and casualties per 100,000 inhabitants. Individuals in isolated cities face a fourfold higher risk of becoming casualties compared with those in more connected cities.

Violence is a pressing global issue that causes nearly half a million deaths worldwide each year[1,2]. Different regions experience distinct forms of violence. For instance, Latin America faces cartel-related and organised crime violence[3], politically motivated groups drive conflict in Africa[4,5], and the United States struggles with mass shootings[6]. Violence has displaced millions, often forcing them to cross international borders[7]. Violence imposes a massive burden on the quality of life, creates lifetime uncertainties, deters investment, and significantly impacts the economy, amounting to nearly 11% of global Gross Domestic Product, while in some countries, such as Syria, Afghanistan, and Iraq, it exceeds 50%[8,9]. Despite differing causes and forms, violent events share common patterns. They are not random but occur under conditions where targets, perpetrators, and enabling factors (like weak state governance) converge[10,11]. These scenarios often repeat, suggesting that one event increases the likelihood of observing subsequent incidents nearby, thus forming spatial and temporal patterns crucial for predicting future violence[12,13].

It is often speculated that violence, across its different forms and expressions, predominantly occurs in urban areas. A weak state in fragile urban settings enables the expansion of non-state armed groups and might drive political violence[14]. Cities aggregate populations, potential targets and victims, resources for armed factions, intra-

elite conflicts, and individuals susceptible to involvement in violence or recruitment by violent groups[5,15]. As a result, cities are often assumed to be inherently violent. Additionally, it has been frequently conjectured that violence has a greater intensity in large metropolitan centres[16]. For example, larger cities have been associated with higher crime rates in some contexts, such as homicides in Brazil and Colombia or burglaries in Denmark and the United States[17,18]. However, other cases, such as homicides in India or burglaries in South Africa and Canada, show the opposite trend[19,20]. Although most of these examples are related to criminal violence, there are also similar challenges regarding political violence, which is also highly heterogeneous and not necessarily an urban issue. Armed conflict, for example, is not mostly urban and has not systematically shifted toward urban areas over time[21]. Moreover, city size does not consistently correlate with terrorist activity or conflict casualties[22]. Instead, violence levels are shaped by additional factors, such as physical geography, proximity, and political dynamics, which often exert a more decisive influence than city size alone[23]. However, it remains an open question whether some forms of political violence are primarily urban and more intense in large cities. This raises a key yet largely unexplored question: how does politically-motivated violence differ across cities of different sizes? What cities are more vulnerable to political violence? What role

[1]Complexity Science Hub, Vienna, Austria. [2]BioComplex Laboratory, Computer Science, University of Exeter, Exeter, UK. [3]Computer Science, Federal University of Ceará, Fortaleza, Brazil. ✉e-mail: r.menezes@exeter.ac.uk

do connectivity and infrastructure play in shaping these patterns, and to what extent do neighbouring locations contribute to regional variations in violence?

Some cities are highly connected, accessible, and central, while others are isolated, remote, vulnerable, and secluded[24,25]. Urban isolation can be interpreted in many ways, such as distance to the capital[26,27] or proximity to an international border, infrastructure and others[28]. Linked to urban connectivity and isolation, it has been suggested that secluded areas rarely offer targets for attacks, and therefore, insurgencies are drawn into central locations[23,29]. Similarly, areas that control access to other places are often strategic targets for groups vying for power and may experience frequent conflict[30]. However, another perspective argues that weaker state presence increases the likelihood of violence, implying that remote areas with limited state capacity face a higher risk of armed conflict[26]. As a result, the question remains open: Are central cities more prone to violence, or do remote, low-accessibility towns face greater risks?

Violence in Africa has become a pressing issue. The continent's murder rate per 100,000 people is five times higher than in Europe and has shown a significant increase over the past decade[2]. Extremist groups have killed thousands and displaced millions, such as Boko Haram in West Africa, which has contributed to the displacement of 2.4 million people[31], and Al-Shabaab in East Africa, which has carried out thousands of deadly attacks over the past 15 years[32]. Countries experiencing ongoing terrorism and insurgencies are also more prone to coups[33]. Since 1950, Africa has accounted for 45% of successful coups (as well as 45% of coup attempts) despite comprising less than 18% of the world's population[33]. Africa's vast geography presents a unique contrast between large metropolitan areas and sparsely populated regions. Megacities such as Cairo, Johannesburg, and Lagos coexist with isolated urban centres that are hundreds of km away from major cities. In these remote areas, limited road infrastructure imposes high mobility costs. Meanwhile, in regions like Greater Cairo, nearly 60 million people fall within its sphere of influence[34]. Growing cities in Africa can both trigger and mitigate violence depending on governance quality, institutional capacity, and their ability to absorb large populations rapidly[35]. At a continental level, Africa provides a diverse and extensive dataset of cities, making it well-suited for robust urban scaling analysis[16]. Many urban scaling studies rely on limited observations, often from wealthy countries (for example, cities in Belgium[20], or in England and Wales[36]), from which "universal" claims are often made[16]. Africa serves as an ideal test case for these urban scaling theories, helping determine whether they hold universally or require reconsideration.

This study examines whether isolated urban areas experience higher levels of political violence and casualties compared to more central and well-connected cities. We use the network of all urban highways to quantify their isolation and consider that cities are "isolated" if they have a limited number of connections or a reduced number of visitors[37]. Analysing political violence at a continental scale presents challenges, particularly in Africa, where data is often scarce and highly susceptible to reporting biases. This is especially true for violence-related data, as many incidents go unreported[38]. To mitigate this issue, we focus specifically on politically motivated violence, for which more granular data is available[39]. While political violence data also carries biases, it allows for more objective comparisons, particularly in terms of long-term trends[40]. By examining the proximity of violent events to urban areas, this study seeks to determine how city size, centrality, and isolation influence the frequency of violence. Utilising data from the Armed Conflict Location & Event Data Project (ACLED)[39], we employ two urban indicators to assess a city's level of isolation: the number of connecting roads and a proxy for intracity travel frequency. The findings reveal that geographical isolation significantly impacts violence. Specifically, individuals in isolated cities experience a casualty rate four times higher than those in centrally

located cities. This underscores isolation not merely as a contributing factor but as a critical determinant of both the prevalence and severity of urban violence.

## Results

To quantify the correlation between city size and violence in Africa, we use the ACLED dataset, which tracks political violence globally through local media reports[39]. The ACLED data does not cover purely criminal activity (so it does not include most types of robberies, domestic violence and other types of crime) but includes gang violence, cartels and terrorist organisations that attack civilians (details in the Supplementary Note 1). ACLED draws on news data, so it is vulnerable to reporting biases[41]. From all the events that occur daily, only a few are deemed relevant enough to be reported as news[42]. Those newsworthy events tend to have specific attributes, so they are biased. For example, in terms of crime, the media usually concentrate on those with a violent component[43]. Thus, the events that are detected by the media (and therefore, the data used here) should be considered a biased subset of all events that occurred in Africa. This is particularly relevant since we aim to compare the number of events and quantify political violence between cities, but the data is likely biased towards detecting more events in specific cities. By quantifying the media attention across cities, it was detected that large urban areas tend to be more newsworthy (meaning that there are more news stories per million people related to events in big cities), so underreporting is more likely in small cities[44].

In Africa, nearly 300,000 politically-motivated events associated with almost 600,000 casualties were reported by the media and registered in ACLED between January 2000 and October 2022[39]. Events are classified into six non-overlapping categories: 25% are classified as battles (interactions between two organised armed groups), 25% as violence against civilians (where an organised armed group deliberately inflicts violence upon unarmed non-combatants), 24% as protests, 11% as riots, 8% as explosions, and 7% as strategic developments (which include non-violent activity by conflict actors, such as arrests and are not considered here). Other datasets, such as the Global Terrorism Database (GTD)[45] and the Uppsala Conflict Data Program (UCDP)[46,47] could also be considered. However, we selected ACLED since it adopts a more comprehensive definition of political violence (including battles, arrests, protests and riots), it has a higher event volume[32] and more frequent updates (see Supplementary Note 1 for details). The data might have systematic reporting biases[38,40]. This is particularly challenging in the case of data that is extracted from the media[48]. Large cities tend to have a stronger media presence, making them focal points for insurgents seeking public and media attention[29].

We then aim to detect how many of those politically-motivated events occurred across different cities and correlate that number with population and other indicators. Although it is possible to use the delineation of cities and use the boundary of each polygon, that process creates a classification using datasets that are not highly accurate at the spatial level, and it is susceptible to how a city is delineated[49]. That test is done in the Supplementary Note 1 and shows results similar to those below. However, this issue could be highly problematic, particularly in more remote and small cities[50]. The precise location of most events is not known, and they are georeferenced in the same location, referring only to a coarse area (see Supplementary Note 1 for two specific cases in Africa). Due to the misalignment between the two spatial datasets, we measure the distance here between the event's location and the centre of the nearest city. Events are assigned to the closest city if they occur within a distance $\delta$ in km of its centre using refs. [51,52]. Then, the number of events and deaths is counted for each city. With this technique, we reduce the prominence of city delineation and count events based on proximity. We detect that half of the violence against civilians and 47% of the casualties occurred within

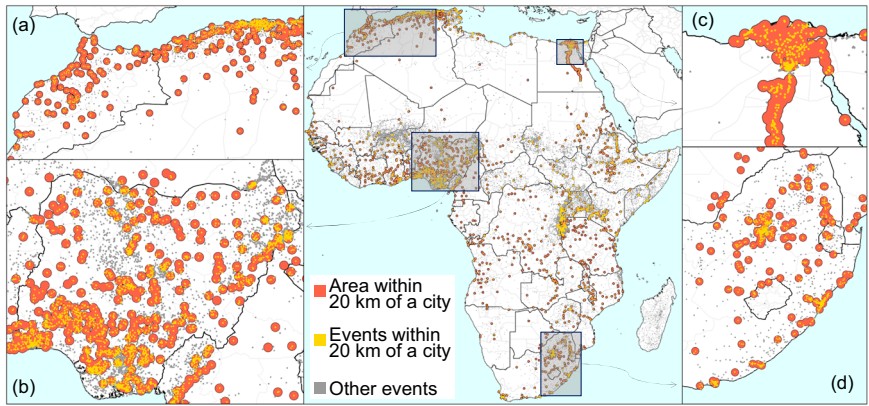

**Fig. 1 | Cities in Africa and politically-motivated events between 2000 and 2023.** Urban Centres and Event Proximity in Africa, with a focus on **a** Morocco and Algeria, **b** Nigeria, **c** Egypt, and **d** South Africa. Most events in Africa occur within a short distance of city centres. This study maps the locations of the largest 2100 cities on the continent, based on data from Africapolis, and identifies all reported events (in yellow) within a 20 km radius of any urban centre. The data used in this figure were sourced from Africapolis.org (OECD/SWAC (2024), Africapolis (database), www.africapolis.org, accessed 2024/07/23) and from ACLEDdata.com (ACLED (2025), Acled (database), acleddata.com, accessed 2022/10/25). Events depicted in grey are situated further away from urban centres. The areas within 20 km of a city centre are marked in red, illustrating the spatial distribution of events in proximity to urban locales.

$\delta = 20$ km of the centre of an African city (Supplementary Note 2). Yet, Africa is a vast territory, where less than 8% of its surface is within 20 km of the centre of a city. Thus, the events detected by ACLED are mostly urban (Fig. 1). Further, results show that the number of events per city is highly heterogeneous and concentrated in a few observations (as observed with violent crime and other events elsewhere[53]). Since 2000, the 5% most violent cities in Africa have had 73% of the fatalities related to the events on the continent, but they are where only 15% of Africa's urban population is currently living. In contrast, nearly two-thirds of African cities have less than one casualty registered yearly related to this type of event. Yet, the high concentration of events and fatalities across African cities does not correspond to the concentration of the population.

The correlation between city size and violence is assessed by fitting Eq. (1),

$$V_i = \alpha P_i^{\beta}, \qquad (1)$$

where $V_i$ is some metric of violence (for example, the number of events or casualties), $P_i$ is the population of the city $i$, and where $\alpha$ and $\beta$ are the model parameters. Values of $\beta > 1$ suggest that people in larger cities suffer more violence. It is frequently assumed that large cities have more violence than small cities. This assumption is often called 'universal', often with a fixed scaling coefficient of approximately 1.15, and is assumed as the result of the increased number of interactions or social activity or increasing costs for providing security in big cities[16,54–56]. For example, it has been found that $\beta = 1.26$ in the case of theft in Mexico, and $\beta = 1.16$ in the case of serious crime in the United States[20], among others (Fig. 2). However, many exceptions have been found, such as burglary in South Africa ($\beta = 0.91$)[20], and the number of crimes ($\beta = 0.8699$) and homicides ($\beta = 0.7788$) in India[19] (an extensive list of scaling coefficients related to violence in the Supplementary Note 3).

In general, although more events and more violence against civilians are registered in larger cities, once considering their population size, we conclude that they are more present and tend to be more lethal in smaller African cities (Fig. 3). The scaling coefficients for the number of casualties, $\beta_L(\delta)$, and the fatalities related to violence, $\beta_V(\delta)$, are both below 1, in fact, below 0.5 for all values of $\delta$ (details in the 'Methods' section). For instance, it is observed that $\beta_L(20) = 0.4865 \pm 0.0013$ and $\beta_V(20) = 0.3404 \pm 0.0025$, indicating that larger cities report fewer casualties per 100,000 inhabitants compared to smaller cities (Supplementary Note 4). This trend persists across

different $\delta$ values. For instance, for $\delta = 10$ km we get that $\beta_L(10) = 0.4954 \pm 0.0014$ and $\beta_V(10) = 0.33349 \pm 0.0029$. When observing the correlations, it is possible that specific cities with a huge population, such as Lagos or Cairo, create a bias in the scaling coefficients[49]. However, we test whether dropping those large cities (known as dragon-kings) or taking randomly generated samples of cities affects the coefficients[57]. We find that the scaling coefficients obtained are not the result of a few large cities among the observations, but we observed a generalised sublinear correlation (Supplementary Note 5).

Political violence has become increasingly lethal in smaller African cities, particularly in recent years. Analysing casualties from the year 2000, the scaling coefficient stands at $\beta_L = 0.4948$, while for violence against civilians, it is $\beta_V = 0.3369$. However, for events occurring after 2015, these coefficients shift to $\beta_L = 0.3431$ and $\beta_V = 0.1739$, respectively (Fig. 3). Restricting the analysis to cover only events between 2020 and 2022 reveals $\beta_L = 0.1851$ and $\beta_V = 0.0308$, suggesting an increasing trend of violence moving away from larger cities (Supplementary Notes 4–6). We find that violence in Africa is sublinear, meaning that small cities have more violence per capita than large cities. Thus, the 'universality' of scaling laws for crime (as discussed elsewhere[58]) applies only to certain countries and types of violence, but not in Africa for battles, violence against civilians, remote violence, riots or protests (Supplementary Note 9).

## Violence is more prominent in isolated cities

The 10% most populous cities of Africa have 66% of the population but only 33% of the fatalities related to politically-motivated violence over the past 22 years. Thus, big cities are not inherently violent, particularly compared to other cities that concentrate higher levels of violence: isolated cities. City connectivity explains the emergence of dominant cities and their economic development and innovation patterns[59,60]. Isolation is one of the main contributors to poverty and a violence generator[61]. Intercity connectivity has been used to improve scaling models[62]. In China, for example, the number of patents in a city is better explained by a city's mobility network than its size[62]. Data corresponding to the location of major highways in Africa from OpenStreetMap was used to measure their level of isolation[63,64]. The degree of city $j$, expressed as $D_j$, is the number of highways that connect that city to others[37]. We classify cities into three groups based on their degree, ensuring each group has roughly the same population (Fig. 4). Cities are labelled as 'high isolation' if $D_j \leq 2$, 'medium isolation' if $D_j \in [3, 5]$, and 'low isolation' if $D_i \geq 6$. Although most cities are highly

isolated, they tend to be small (Supplementary Note 7). In cities characterised by high isolation, the rate of registered casualties stands at 3.9 per 100,000 people annually, in stark contrast to just 0.7 per 100,000 in cities with low isolation levels. This stark discrepancy underscores that lethality in highly isolated cities is 5.4 times higher than in their less isolated counterparts (Fig. 3a). Specifically, when considering violence against civilians, the lethality rate in cities with high isolation escalates to 7.4 times that of more centrally located cities.

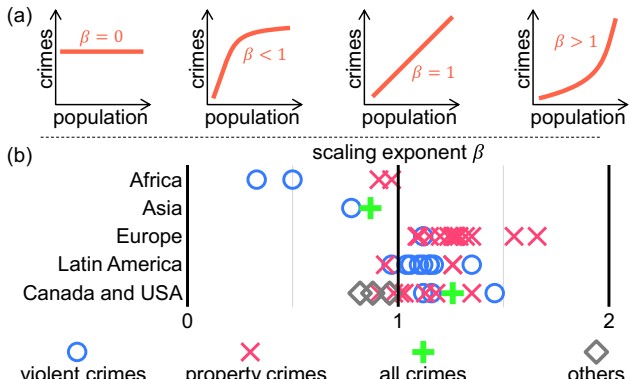

**Fig. 2 | Coefficients associated with the urban scaling of violence. a** Schematic impact on the number of crimes as a function of city size, following a power law $V_i = \alpha P_i^\beta$ with different values of $\beta$. For $\beta = 0$, all cities have the same number of crimes regardless of their population. For $\beta \in (0, 1)$, crime scales sublinearly with population; for $\beta = 1$, it scales linearly; and for $\beta > 1$, it scales superlinearly. **b** Scaling coefficients across the world. Observed scaling coefficients for different types of crime and violence suffered in some parts of the world. Coefficients obtained for Brazil, Canada, Colombia, England and Wales, Italy, Mexico, the United States and the United Kingdom.

The degree does not capture a city's relationship with the rest of the network but only with its adjacent locations. Thus, a second metric is considered based on the estimated number of modelled journeys that pass through each city. The motivation behind this is to detect regions with a weaker connectivity to the rest of the urban system and quantify if they have higher levels of violence[28]. Other metrics have also been proposed, for example, the travel time to the capital[26]. However, that method ignores cross-border interactions and urban corridors (such as the corridor Lagos–Accra) and changes substantially if we consider the political capital of a country or the biggest city (like Yamoussoukro, the capital of the Ivory Coast or Abidjan, its largest city with 30 times more population). The number of journeys is calculated by assigning trips generated between pairs of cities using a gravity model that considers the fastest route. The centrality of city $j$, expressed as $C_j$, is the number of trips that travel through that city[37]. It is a form of weighted betweenness which captures the level of centrality of a city based on its size and distance decay effects (details in the Methods).

Cities with a small centrality $C_j$ are locations with only a few journeys, which tend to be far from big cities and away from significant corridors (Supplementary Note 7). In cities ranking in the lowest 25% for centrality, lethality rates are 15.1 times higher than those in the top 25% for centrality, highlighting that remote and peripheral cities exhibit significantly greater lethality compared to their more central counterparts. This disparity extends to violence against civilians, which shows a similar pattern with a ratio of 15.8. Our analysis indicates that remote and isolated locations tend to have higher levels of violence. Additionally, at the country level, lower income is also correlated with a higher number of events and fatalities (Supplementary Note 8). Notably, ACLED events are more present in small and isolated cities, and that has intensified over the past two decades (Fig. 3c). Isolation reduces the capacity of the state to react and allows extremist groups to attack and defend their territory[30]. Here, we have also

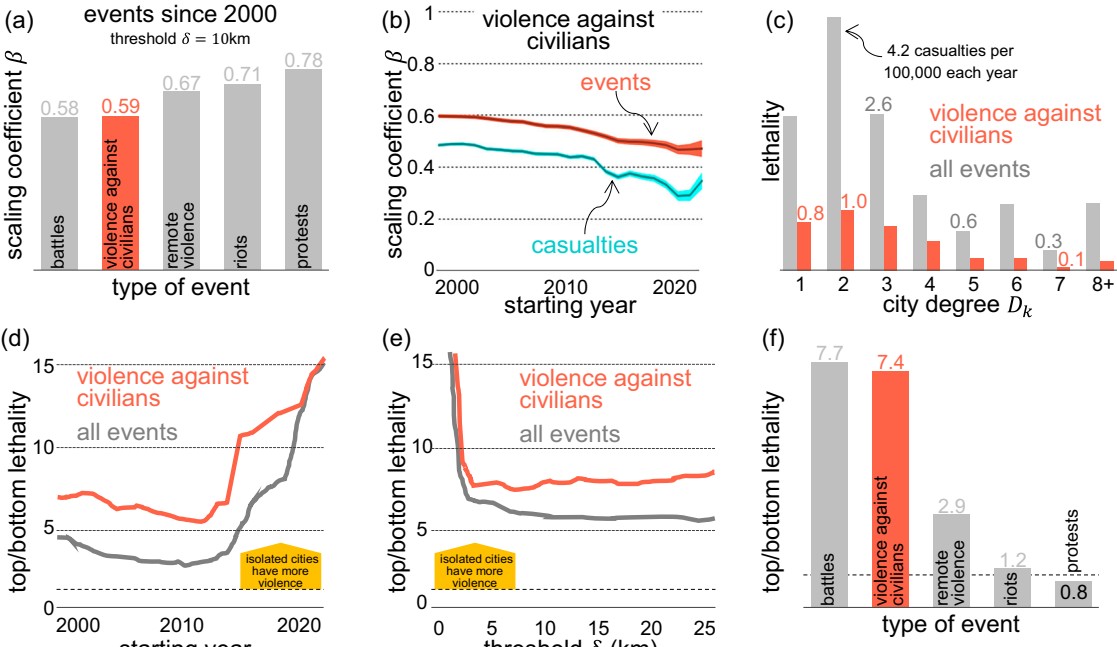

**Fig. 3 | Analysis of urban lethality dynamics. a** Scaling coefficient observed for different types of events registered in ACLED since 2000 with a distance threshold $\delta = 10$ km. **b** Scaling coefficient of violence against civilians suffered across cities of varying sizes by taking the events starting from a varying year. **c** Lethality (vertical axis) across cities of varying connectivity (horizontal axis). **d** Disparity in lethality (vertical axis) over time, considering events from specific years (horizontal axis). **e** Comparison of highest vs. lowest lethality (vertical axis) based on the distance threshold $\delta$ (horizontal axis) applied to attribute events to cities. **f** Lethality contrast (vertical axis) by type of event, merging 'protests and riots' as well as 'battles and strategic developments' into unified categories from the ACLED database. The dashed line indicates a lethality rate of $\phi = 1$, the hypothetical rate if the most isolated cities experienced the same lethality as the most central cities.

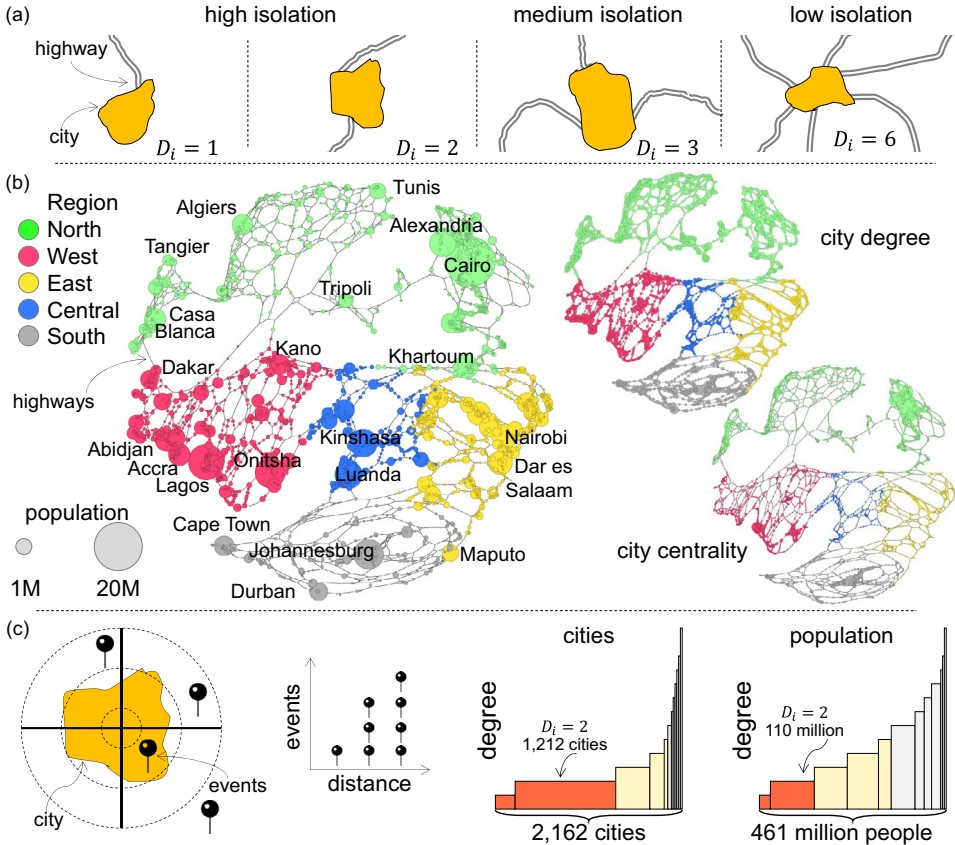

**Fig. 4 | Politically-motivated violence and urban isolation in Africa. a** Based on the road network in Africa, cities are classified by their degree and by their centrality. **b** The network formed by African cities and their highways enables us to classify cities based on their population size, as well as their degree and centrality, which is a form of weighted node betweenness. **c** Most urban areas exhibit a high level of isolation ($D_j \leq 2$), predominantly consisting of smaller cities. Among the 2162 cities analysed in Africa, 67.1% are highly isolated, yet these cities comprise only 29.7% of the considered urban population. Conversely, cities with low isolation constitute merely 3.7% of the total, but they represent 28.1% of the urban population.

---

detected that isolation is correlated with a higher number of battles between politically-motivated groups, more violence against civilians and more fatalities as a result.

## Discussion

In African urban landscapes, larger cities report a lower incidence of battles between politically motivated groups, a lower rate of violence against civilians and fewer riots and protests compared to their smaller counterparts. Despite the higher absolute casualty numbers in larger cities, individuals in smaller cities face a significantly increased risk of politically-motivated violence. This heightened vulnerability extends beyond urban dimensions, particularly affecting small, poorly connected urban centres. These areas frequently become targets for ethnic militias, rebel factions, and Jihadist groups, often outpacing the response capabilities of state forces[5]. The challenge of isolation complicates the situation further, impeding economic growth, elevating the cost of goods and services, diminishing resilience, and restricting access to fundamental services like healthcare and education.

This work used a static observation of cities (data from Africapolis[65]) and of the infrastructure that connects them (data from OpenStreetMap[37,64]). However, more than 20 years of events from ACLED were used to assess the violence in cities[39]. Consequently, results for earlier years must be approached with caution, as cities are rapidly growing, and new agglomerations are also emerging. Nevertheless, in recent years, isolated cities have experienced violence ten or more times more frequently compared to central cities. Hence, even if the observed trend could be influenced by the emergence of new cities

in 2021 and 2022, people in isolated cities suffered 14.5 times more casualties related to violence against civilians.

Events were initially assigned to cities by examining their exact locations and the delineation of urban polygons, followed by proximity to the centre. Both techniques have technical issues. Firstly, the location registered for the events might not be precise, especially if they occur in remote areas with fewer reference points to accurately capture them (Supplementary Note 1). Secondly, small cities have a footprint of only a few square kilometres, whereas major cities occupy larger areas. However, a wide range of distances was tested to match events to cities, and it was found that the results hold, regardless of the length (Supplementary Note 2). Some events do not need to occur precisely inside the polygon of a city for the population to suffer their consequences. Thus, even if the spatial match between events and cities is not perfect, it is more pertinent to detect who suffers violence rather than the precision of an event being registered within a town or a few kilometres.

The dataset might not capture some events, particularly if they were not reported in local media. Although it is possible because the media cannot keep up with reporting minor events in big cities, here we are considering events related to violence against civilians, which cause, on average, 2.5 casualties each. Thus, these events tend to be very visible. In fact, incidents in large cities are usually better documented by the media than in small locations[44]. Therefore, it is likely that violence against civilians is even more present in small and isolated cities than the results suggest.

The incidence of violence on the African continent has markedly increased, with isolated cities becoming increasingly vulnerable. From

2010 to 2021, the number of fatalities resulting from violence against civilians escalated dramatically, rising from under 4000 to nearly 16,000 annually, a 340% increase. This sharp increase has happened primarily in remote areas. These isolated cities provide strategic advantages for some groups, exploiting the sparse surveillance of limited road networks to establish safe havens[66]. Even basic security measures, such as deploying a few lookouts, can delay state intervention, offering these groups considerable lead time. Addressing this surge in violence necessitates substantial investments in infrastructure and connectivity, along with efforts to strengthen national cohesion.

## Methods

### Cities and their level of isolation

Data from Africapolis provides the location and population estimates for all cities considered[65]. A critical aspect of the analysis of cities is delineating them, as defining their boundary is not trivial, and different criteria produce distinct and often contrasting results[49]. However, in Africapolis, cities were identified using census data and satellite imagery, applying the same definition across the continent. Therefore, it is feasible to analyse data at the continental level. The network of African highways was constructed from Africapolis cities and with data from OpenStreetMap[63]. It consists of two datasets: edges and nodes. All cities with more than 100,000 inhabitants and smaller towns near main highways are included in the cities dataset. We consider 2162 cities totalling 460 million inhabitants; roughly half of the continent's population. The 9159 edges correspond to existing highways on the continent, representing roads connecting cities and road intersections in Africa[37,64]. For each edge, an estimate of the time required to travel through that edge is included, aiming to capture the road's straightness, the quality of the road, and other infrastructure attributes. The 7361 nodes in the network correspond to either cities (2162) or road crossings (5199).

The degree of city $j$, expressed as $D_j$, corresponds to the number of highways that connect that city to others. Thus, a city with degree $D_j = 1$ has only one road to travel from and to that city, whereas a city with degree $D_j = 2$ is an urban agglomeration growing around a main highway (the highway goes through the city). The degree does not capture a city's relationship with the rest of the network but only with its adjacent nodes. That is, the city $j$ could have degree $D_j = 2$, for example, but one of its neighbouring nodes could be another city with degree $D_k = 1$, thus suggesting that both cities, $j$ and $k$, are very isolated, and people will rarely travel to or through them. However, a city with $D_j = 2$ could be adjacent to two large cities on each side, suggesting that it is more central in the network and people are likely to travel more frequently through it. Hence, a second metric trying to capture this nuance is the centrality of a city based on the estimated number of journeys that pass through it. Although one option is to consider the node betweenness directly from the network, it is crucial to consider two aspects of cities. First, city size is highly skewed, meaning that large cities such as Cairo or Lagos are thousands of times bigger than small cities. Second, long-distance journeys are not as frequent as short ones. Therefore, instead of the node betweenness, a weighted betweenness is considered, aiming to capture the level of centrality of a city. The purpose of this indicator is to estimate the number of journeys travelling through each city. The flow between two cities is estimated using a gravity model. Gravity models are frequently used to analyse spatial interactions, capturing size and distance[67,68]. In its simplest form, the flow $F_{o,d}$ between the origin city $o$ and the destination $d$ is estimated by

$$F_{o,d} = \frac{P_o P_d}{N_{o,d}^\gamma},\tag{2}$$

where $P_o$ and $P_d$ are the population sizes of the origin and destination, respectively, $N_{o,d}$ represents the travel time, and $\gamma \geq 0$ captures the

impact of travel time on flow. International borders create significant delays and travel frictions (particularly in some parts of Africa[25,69]), so here we use an estimate for the travel time across different types of roads, and we add two hours for each border crossing, representing the cost that borders impose on the intermediacy of cities and countries[37]. The value of $\gamma = 2.8$ has been utilised previously, signalling high travel frictions on the continent[37]. Subsequently, the estimated flow is assigned to the network through the fastest path. Finally, centrality is defined as the number of trips that travel through city $j$, expressed as $C_j$. It is estimated by summing the flow that passes through each node in the network when considering all pairs of cities. Formally,

$$C_j = \sum_{o,d} F_{o,d} H_{o,d}(j),\tag{3}$$

where $H_{o,d}(j) = 1$ if city $j$ is on the route between $o$ and $d$, and zero otherwise. Cities with a small centrality $C_j$ have few journeys passing through them and, therefore, exhibit high-level isolation. Cities with a high level of isolation tend to have a small degree, but also tend to be far from large cities and away from commercial corridors (Supplementary Note 7).

### Identifying urban events and comparing their lethality

The ACLED is the most comprehensive and detailed database available for analysing violence in Africa[39]. Although there are other sources to analyse conflict, the ACLED database stands out as an event-based database, detailing the location and estimated number of fatalities for each event (more details in the Supplementary Note 1). ACLED categorises six types of events: battles, violence against civilians, explosions/remote violence, strategic developments, riots, and protests. In this analysis, riots and protests are also considered, although they are not typically caused by politically-motivated violent groups (a more detailed analysis of protests and riots is in the Supplementary Note 9). From January 2000 to October 2022, ACLED reported more than 182,000 events in Africa associated with politically-motivated violent groups, resulting in more than 560,000 deaths (Table 1).

Each event is assigned to a city based on the distance between the event and cities nearby. To assign events to cities, we consider the distance between the location of events and the centre of cities. For event $i$, we measure the distance in km to the centre of its nearest city $d_{ij}$ and then assign it to the nearest city $j$ if $d_{ij} < \delta$, for some threshold $\delta > 0$. We will vary the values of $\delta$ between 1 and 30 km to test and ensure that the results do not depend on how events are assigned to cities. The rationale behind this method is that even if violent events do not occur precisely within an urban polygon, they take place within its outskirts, affecting the nearest urban population. Events at a distance larger than $\delta$ from all cities are classified as rural (Supplementary Note 2).

The impact of events is vastly heterogeneous. Some events are major incidents with hundreds of casualties, while others are much more minor. To compare the lethality across cities, we consider all events that were assigned to the city $j$ and define the lethality as

$$\phi_\delta(j) = \frac{\sum_{i \in I_\delta(i,j)} f_i}{P_j},\tag{4}$$

where $P_j$ is the population of city $j$, and $I_\delta(i,j) = 1$ if event $i$ was assigned to city $j$ and zero otherwise, and $f_i$ is the number of fatalities of event $i$. Thus, Eq. (4) gives the number of casualties assigned to the city $j$ divided by its population (with numbers reported in terms of 100,000 inhabitants). The lethality of city $j$ is a comparable metric across cities that gives the combined impact of politically-motivated violence in each urban area.

**Table 1 | The ACLED dataset has registered more than 180,000 events with more than 560,000 casualties in Africa since the year 2000 related to battles, violence against civilians, explosions and remote violence, and strategic developments**

| Type of event | Number of events (2000–2022) | Number of casualties (2000–2022) |
|---|---|---|
| Battles | 71,323 | 317,795 |
| Violence against civilians | 70,726 | 182,875 |
| Explosions and remote violence | 21,852 | 59,284 |
| Strategic developments | 19,027 | 533 |
| Total | 182,928 | 560,487 |

Expressing the number of casualties in city $i$ as $L_i(\delta)$ and those related only to violence against civilians as $V_i(\delta)$, we explore the scaling relations $L_i \sim P_i^{\beta_L(\delta)}$ and $V_i \sim P_i^{\beta_V(\delta)}$. A scaling relationship of an indicator $Y$ with a coefficient $\beta > 1$ is termed superlinear, indicating that larger cities have disproportionately more of $Y$ than smaller cities. With $\beta \approx 1$, the size of a city has little impact on the distribution of $Y$; with $\beta < 1$, smaller cities have more $Y$, per capita, than larger cities. For instance, for serious crimes in cities in the United States, it was observed that $\beta_S = 1.16 \pm 0.04$, suggesting that larger US cities experience more serious crimes per capita[58]. Here, we investigate whether the occurrence of casualties and violence against civilians increases in larger cities.

Further, we aim to determine if cities with high isolation exhibit, as a group, greater levels of violence. We construct a metric analogous to the one defined in Eq. (4) but applied to a group of cities, such as those identified with high isolation. For a group of cities $J$, their collective lethality is defined as

$$\phi_\delta^{(J)} = \frac{\sum_{j \in J} \sum_{i \in I_\delta(i,j)} f_i}{\sum_{j \in J} P_j}. \tag{5}$$

This equation calculates the total number of casualties across all cities in group $J$, divided by their combined population size. With $I$ representing cities with high isolation and $L$ those with low isolation, we find that $\phi_{10}^{(I)} = 0.99$ and $\phi_{10}^{(L)} = 0.13$, indicating that cities with high isolation are eight times more violent than those with low isolation.

**Comparing cities of different levels of isolation**
We define the isolation impact $\theta_\delta$ as the ratio between the lethality of cities with high isolation and those with low isolation. If violence were not associated with the degree of isolation of a city, we would expect to see values of $\theta_\delta \approx 1$ for certain values of $\delta$. However, this is not observed, and $\theta_\delta > 1$ for all values of $\delta$ (Fig. 3). Similarly, the centrality impact is defined as $\theta_\delta^c$ by distinguishing the 25% of cities with the lowest centrality from the 25% with the highest centrality and calculating the ratio of their lethality. If centrality did not influence the distribution of violence, $\theta_\delta^c \approx 1$ would be expected; yet, this is not the case.

Some events result in a high number of fatalities, while most report none. The top 1% of most lethal events account for nearly 40% of the casualties documented by ACLED in Africa. A critical examination involves assessing whether the impacts of isolation or centrality stem from just a few high-impact events in a limited number of cities. To test (and reject) this hypothesis, a subset of events is sampled, constructing the same metrics for analysis. Sampling half of the reported events and measuring the impact of isolation and centrality for this subset, the process is repeated 1000 times to observe potential fluctuations from considering only some events. If the impact of isolation or centrality were due to a few events in a few cities, these events would eventually be excluded, leading to $\theta_\delta^{(s)} \approx 1$ in some iterations. However, across all $\delta$ values and sampling iterations, $\theta_\delta$ and $\theta_\delta^c$ significantly exceed one, affirming that isolated cities consistently experience higher violence than central cities.

Assigning events to cities depends on the parameter $\delta > 0$, considered within the [1, 30] km range. For smaller cities, lower $\delta$ values suffice to encompass their area, whereas larger cities require higher values to include urban and most peri-urban events. With $\delta = 30$ km, events within this radius are attributed to the city's population. Despite the dependency on $\delta$, isolated cities consistently exhibit higher lethality across all values of this parameter. Highly isolated cities are found to be at least five times more lethal than central cities, with the ratio rising to seven when focusing solely on violence against civilians (Supplementary Note 2). By halving the event sample and recalculating $\phi_\delta$, the observed lethality ratio for isolated versus central cities remains significant across all $\delta$ thresholds and sampling iterations (Supplementary Note 5). Thus, the heightened violence in isolated areas is not merely a result of event assignment or the presence of a few high-impact incidents but reflects a broader pattern where isolation, coupled with reduced state presence, elevates violence risk.

**Reporting summary**
Further information on research design is available in the Nature Portfolio Reporting Summary linked to this article.

## Data availability
The data for all African urban agglomerations, Africapolis, are available at the OECD website https://africapolis.org/en/data. The data corresponding to the African urban network are available here https://github.com/rafaelprietocuriel/AfricanUrbanNetwork. The data corresponding to all events in the continent, ACLED, are available here https://acleddata.com/.

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

## Acknowledgements
This work was funded by the Austrian Federal Ministry for Climate Action, Environment, Energy, Mobility, Innovation and Technology (2021-0.664.668)—R.P.C., and by the Austrian Ministry for Innovation, Mobility and Infrastructure (GZ 2023-0.841.266)—R.P.C.

## Author contributions
R.P.C. conceived the study and the methodology. R.P.C. and R.M. analysed the results and wrote the manuscript.

## Competing interests
The authors declare no competing interests.
