## [Transparent Peer Review file · Nature Communications]

Violence, City Size and Geographical Isolation in African Cities

Corresponding Author: Dr Rafael Prieto-Curiel

Version 0:

Reviewer comments:

Reviewer #1

(Remarks to the Author)

This is an interesting manuscript, revisiting some claims in the literature on the scaling of violence in cities. Here, the authors investigate the statistical relation between violence and city size in African countries. They observe that, contrary to previous claims in the literature, the relation is very different from what was observed in other countries and continents, thus questioning the so-called "universality" of the scaling. This contribution is an interesting contribution to the literature, but I would say a fairly incremental one, which will be of interest to a more specialised audience.

The main contribution of the work is their investigation of the relation between the location of a city and its level of violence. This is an interesting research question, but I am not convinced that the analysis provides sufficient support for the claims. As the title writes "Geographical Isolation Amplifies Political Violence in African Cities", the authors search to show that isolation causes political violence. At best, though, they observe a statistical relation but it is unclear if there is any causal relation. The relation could be due to other factors (e.g. socio-economical development is expected to be related to connectivity, as the authors note). Given the observational nature of the work, it is unclear to me that the authors will be able to support this claim. Yet, I would encourage them to perform additional analysis, for instance by including socio-economic indices of cities in the analysis. In any case, some of the claims should be toned down.

Reviewer #2

(Remarks to the Author)

The manuscript covers an interesting and relevant topic, and presents a sophisticated analysis of geographic patterns of violence. It also connects to a broad relevant literature on urban violence, situating the study within this body of research in a good way. I find the shift to focus on violence per capita (i.e., how likely is a person living in a given city to experience violence) well motivated. However, I see some major issues with the manuscript in current form, and cannot recommend publication. I outline my key concerns below, and hope that they are helpful to the authors in developing the manuscript.

The most important problems I see is that the argument is theoretically underdeveloped, and that the manuscript makes some rather strong claims (about problems in previous research, and about the findings of the study) while overlooking data and identification problems that complicate these conclusions.

A first point is that the concept of violence is used without clear conceptual definition. In the introductory section, "violence" is extremely broad. Do you include eg domestic violence? The references in this section are mainly about political and gang violence.

Situating the argument, you say on p. 1 that "Although rural and sparsely-populated areas might be perceived as conducive to insurgencies due to limited state reach, they rarely offer a rich set of targets for violent activities. Violence mainly manifests as an urban phenomenon." Firstly, is it really proven that "violence" mainly takes place in cities? A significant literature has questioned the ability to draw such conclusions, given the systematic reporting biases that make urban violence more likely to get reported (see e.g. Dietrich & Eck 2020; Von Borzyskowski & Wahman 2021; Weidmann 2015). Secondly, rural areas are not "empty" of possible targets; they contain populations, security infrastructure, symbolic cultural sites, etc. Clarify what you mean here.

The argument/relationship under study is not fleshed out theoretically. To understand why isolated cities should be at higher

risk of violence, the argument about targeting necessitates clarification on exactly what form(s) of violence you include. Is it specifically violence by organized actors with political motives? At the end of the paper you clarify that "riots and protests are not considered, as they are not typically caused by politically-motivated violent groups" (aren't they?) – this information should come much earlier, and be explicitly theorized. It seems to imply that your argument is much more limited than the front-end of the manuscript suggests. Is your argument specifically about violence within a broader context of ongoing civil war? It also appears that testing the argument could be difficult due to possible reverse causality, whereby cities located in contested regions of a state also become more isolated.

The research design is in many ways impressive, but as a reader I want a lot more critical discussion about the data used (both on violence, and on cities), the possible biases and limitations associated with that data, and the implications for your analysis and results. ACLED draws on news data and is as such vulnerable to reporting biases (see the sources referenced below; in addition, Eck 2012). More specific to the argument presented here, is it possible that in larger cities, "everyday"/"routine" violence (e.g. criminal violence in slum settlements) goes unreported to a higher extent? When it comes to the city population data, how reliable is it? See e.g. Satterthwaite 2010 for a critical discussion.

Your empirical focus on Africa is not motivated or discussed. Do you expect relationships between geographic isolation and violence to be different in Africa than other regions? Are there important scope conditions for your argument? Discussing the connectivity component of the analysis, on page 5 you note that "city connectivity explains the emergence of dominant cities and their economic development and innovation patterns. Isolation is one of the main contributors to poverty and a violence generator." So is it mainly a grievance argument?

In sum, the manuscript makes a strong causal claim, but neither theorizes a clear causal argument nor presents a convincing causal identification strategy.

Smaller comments

Discussing previous research on city size and violence (p. 2), you note contradictory findings, and conclude that "city size has been wrongfully assumed to capture violence heterogeneities, and large cities have also been wrongfully deemed to be more violent than small cities." This is a strong claim; if taking the analytical approach in your paper (focusing on per capita levels), it seems some of the research points in that direction, but overall the current state of knowledge cannot be summed up in that claim.

On p. 4, you note that "The 5% most violent cities in Africa had 73% of the fatalities related to the events in the continent since 2000 but is where only 15% of Africa's urban population is currently living." This makes me wonder if there are a few key cases that drive the results to high extent (eg a few highly violent cities in the context of Ethiopia's civil wars)? Show the reader an overview of the most affected cities, and discuss (and analyse) the impact of outlier cases.

I like the careful operationalisation of connectivity and believe it is a key contribution of the paper. However, is it not likely to capture remoteness in roughly the same way as previous work relying on travel time to capital, proximity to borders (being on the "edge of the state") etc? More discussion and reference to this literature (e.g. Fearon & Laitin and Müller-Crepon et al that you cite earlier; perhaps also Buhaug 2010), explaining how your operationalisation differs, would be useful.

References

Buhaug, Halvard. "Dude, where's my conflict? LSG, relative strength, and the location of civil war." *Conflict Management and Peace Science* 27.2 (2010): 107-128.

Dietrich, Nick, and Kristine Eck. "Known unknowns: media bias in the reporting of political violence." *International Interactions* 46.6 (2020): 1043-1060.

Eck, Kristine. "In data we trust? A comparison of UCDP GED and ACLED conflict events datasets." *Cooperation and Conflict* 47.1 (2012): 124-141.

Satterthwaite, David. "Urban myths and the mis-use of data that underpin them." In: Beall, Guha-Khasnobis and Kanbur (eds., 2010) *Urbanization and development: multidisciplinary perspectives*. Oxford University Press.

Von Borzyskowski, Inken, and Michael Wahman. "Systematic measurement error in election violence data: Causes and consequences." *British Journal of Political Science* 51.1 (2021): 230-252.

Weidmann, Nils B. "On the accuracy of media-based conflict event data." *Journal of Conflict Resolution* 59.6 (2015): 1129-1149.

Version 1:

Reviewer comments:

Reviewer #1

(Remarks to the Author)

The authors have addressed my comments.

Reviewer #2

(Remarks to the Author)

As I noted earlier, this is a study on an interesting and relevant topic, featuring a novel and thorough analysis of empirical patterns. In the revised manuscript, the authors have addressed many of my key concerns about causal inference and the theoretical argument. Notably, the focus on Africa is well motivated (and it is clear that there is an empirical research gap here). However, I still see a few important areas that need to be unpacked or clarified before the manuscript is suitable for publication.

First, while the authors have clarified the discussion of violence in parts, it still remains unclear to the reader at the outset of the manuscript what forms of violence are in focus. In the first paragraph, the authors speak of 'violence' in a very broad sense (and make rather sweeping claims about how different forms of violence characterize different regions). It is not clear to the reader if claims about violence being an urban phenomenon encompasses all these form of violence. Importantly, the literature most strongly connected to this claim is the criminological literature (the references cited on page 4 in connection to the 'universal assumption', and most or all of the works producing scaling coefficients in SM-C are about criminal violence). Either the authors need to back up if and how this assumption has also been made with reference to political violence, or they should explicitly review literature on the urban/rural dynamics of political violence to situate their study and contribution in that literature.

In addition to clarifying the contribution, the authors should be clear on what form(s) of political violence are in focus in the empirics. The emphasis shifts in different places, from political violence broadly (e.g. page 3) to violence against civilians specifically (e.g. in the abstract). In the methods section (p. 3), it sounds like all the categories of ACLED violence are included except protests and riots. However, when discussing the results, you focus on violence against civilians. Do you zoom in on this category as a sub-analysis, and if so, why? Motivate this upfront.

I also believe more critical discussion of the data is needed. The authors have included discussion of the possible news biases, which is good. However, another key dimension is coding precision, which only comes up as part of the appendix. The authors have included Eck's (2012) critical comparison of ACLED and UCDP but only cursively engage with it (she reports quite large quality issues with ACLED, notably regarding subnational dynamics of violence). These issues may have improved since, but at the least, the authors should motivate why they prefer ACLED over other georeferenced violence datasets, and critically discuss possible limitations. Importantly, how many of the events are precisely coded (in terms of geoprecision) and could this affect patterns across central and more marginalized areas? In the codebook, ACLED notes that "If the source material indicates that activity took place in a small part of a region, and mentions a general area, the event is coded to a town with geo-referenced coordinates to represent that area, and the 'Geo-precision' code 2 is recorded" (p. 36). How frequent is this type of precision code, and could it affect the results (if e.g. many events in marginalised regions are coded to the nearest town)?

Version 2:

Reviewer comments:

Reviewer #2

(Remarks to the Author)

The authors have addressed all my remaining concerns in a convincing and transparent way. I recommend the revised manuscript for publication.

REVIEWER 1

***Reviewer (R1.1):** This is an interesting manuscript, revisiting some claims in the literature on the scaling of violence in cities. Here, the authors investigate the statistical relationship between violence and city size in African countries. They observe that, contrary to previous claims in the literature, the relation is very different from what was observed in other countries and continents, thus questioning the so-called “universality” of the scaling. This contribution is an interesting contribution to the literature, but I would say a fairly incremental one, which will be of interest to a more specialised audience.*

Thank you very much for your comments. We appreciate that our claims may be perceived as being directed more for a specialised audience. However, we consider that there are three relevant aspects of our study. Many urban scaling studies have claimed that they have observed some universal pattern, meaning that we have an idea of how small or large cities function. These claims have worked particularly in terms of GDP, patents, infrastructure and social interactions Bettencourt et al. (2007). Yet, some of these claims were based on a biased set of observations, for example, taking only cities in the USA. When we observe other regions in the world, more and more, we notice that cities do not follow those universal patterns. In our study, we contribute to debunking those universality claims, so this is not an incremental study of violence. Additionally, we show that observations related to the scaling of violence cannot be claimed as universal in the USA and Canada. Rather, there is a wide variety of coefficients, where some are sublinear and others superlinear, depending on the type of crime and period considered. One of the relevant claims related to the scaling of violence is that since “crime is superlinear with respect to city size” then large cities are more violent than small cities Bettencourt et al. (2007). However, we observe that large cities are not necessarily more violent than small cities (adjusting for population). Large cities are not condemned to be more violent, as one would assume, based on the scaling studies. This contribution is relevant to those who study cities and analyse scaling results Alves et al. (2013); Schläpfer et al. (2014); Bilal et al. (2021). Examining 54 countries over 20 years is a more rigorous test for universality claims, particularly when compared to research centred on one or two high-income regions.

Our second contribution is related to the studies of violence. We analyse years of violence against civilians in Africa and show that it is correlated to the isolation of cities. We observe that with higher levels of isolation, there is more violence, and that has been observed for years. The absence of state forces and law enforcement causes a gap that is frequently occupied by criminal organisations. Thus, to study

the impact of violence on people who are displaced by it, it is crucial to consider the (sublinear) effect of city size and the costs of isolation.

Finally, in Africa, nearly one-third of the urban population lives with high levels of isolation, and they are the ones at high risk of violence and, thus, displacement due to violence. Africa consists of 54 independent nations, which account for over one-quarter of all countries in the world—each with unique cultural, linguistic, and socio-political characteristics. Indeed, Africa has long been recognised as housing some of the most ethnically diverse countries on the planet. Yet, there is a limited number of studies related to African cities and almost no analysis related to African urban scaling. African cities are the ones that will have the fastest growth in the world in the upcoming decades, and it is where a huge part of the population growth will happen. Thus, the relevance of incorporating African cities in the ongoing scientific debates.

***Reviewer (R1.2):** The main contribution of the work is their investigation of the relation between the location of a city and its level of violence. This is an interesting research question, but I am not convinced that the analysis provides sufficient support for the claims. As the title writes “Geographical Isolation Amplifies Political Violence in African Cities”, the authors seek to show that isolation causes political violence. At best, though, they observe a statistical relation, but it is unclear if there is any causal relation. The relation could be due to other factors (e.g. socio-economical development is expected to be related to connectivity, as the authors note). Given the observational nature of the work, it is unclear to me that the authors will be able to support this claim. Yet, I would encourage them to perform additional analysis, for instance, by including socio-economic indices of cities in the analysis. In any case, some of the claims should be toned down.*

We agree with the reviewer that our results are correlational, and we cannot claim any causal relation. Yet, we have two issues related to conducting additional analysis in African cities. First, data at the city level is scarce. There is limited information regarding socioeconomic indices of cities. For example, the UN Economic Commission for Africa (UNECA) has an ongoing initiative to estimate the GDP at the city level. See <https://www.uneca.org/city-gdp-estimation-africa>. The lack of data is a critical issue in understanding Africa, particularly its cities. The most recent census from some African countries dates back decades, such as Congo (1984) or Somalia (1975). Almost no data at the city level exists beyond some estimates (such as the one we used using Africapolis OECD/SWAC (2018)) and our metrics of isolation or city degree.

Second, in the hypothetical scenario where we have data at the city level (for example, GDP per person), we would still have correlations and not any causal mechanism. For instance, we will likely find that isolated cities tend to have lower incomes (due to the lack of industry and higher prices for obtaining goods). Thus, we likely find that cities with a lower GDP per person tend to have higher levels of violence. However, the causal link between isolation, GDP per person and violence would remain unclear. Additionally, those links cannot be tested using our mechanisms. Establishing causality would require either natural experiments (like sudden changes in isolation due to new infrastructure) or quasi-experimental designs that are rarely available in this context. Even with longitudinal data, the slow-changing nature of both isolation and economic development makes it difficult to disentangle their effects. The complex interplay between these factors (where isolation might affect violence both directly and indirectly through economic channels) creates challenges

for identification that standard econometric techniques cannot efficiently resolve. Furthermore, potential instrumental variables that might affect isolation but not violence through other channels are scarce in the African context.

We have, however, expanded our project, first by checking if our results are dependent on a few cities. It was shown that specific cities with huge population sizes, known as dragon-kings, could affect the results when estimating the scaling law parameters since they might display different patterns Arcaute et al. (2015). A method to detect whether results depend on only a few cities is through the analysis of randomly generated samples Cabrera-Arnau and Bishop (2020). With the inclusion or exclusion of dragon kings in the scaling analysis through sampling, the impact of a few cities on the estimated scaling parameters was detected. We have added this analysis to our manuscript by sampling cities in Africa and observing deviations corresponding to dragon-king cities. The results show that the most significant effect of the scaling parameters is still observed, regardless of the sample of cities taken, so results do not rely on a few outliers. We have added the subsection *E - Are the scaling coefficients the impact of only a few cities?* in the Supplementary Materials to expand on this topic and mentioned it shortly in the Results.

Finally, we have also tested the result after controlling for the GDP per person of a country. That indicator is more readily available and reliable when compared among countries. Results of the test are also correlational as it is not possible to disentangle the complex feedback loops between isolation and wealth and between violence and wealth. We added the subsection *H - Income and violence against civilians* in the Supplementary Materials (SM).

REVIEWER 2

***Reviewer (R2.1):** The manuscript covers an interesting and relevant topic, and presents a sophisticated analysis of geographic patterns of violence. It also connects to a broad relevant literature on urban violence, situating the study within this body of research in a good way. I find the shift to focus on violence per capita (i.e., how likely is a person living in a given city to experience violence) well motivated. However, I see some major issues with the manuscript in current form, and I cannot recommend publication. I outline my key concerns below, and hope that they are helpful to the authors in developing the manuscript. The most important problem I see is that the argument is theoretically underdeveloped, and that the manuscript makes some rather strong claims (about problems in previous research, and about the findings of the study) while overlooking data and identification problems that complicate these conclusions.*

Thank you for your detailed and thoughtful feedback on the manuscript. We greatly appreciate your acknowledgement of the relevance of the topic and the sophistication of our analysis, as well as your recognition of how the study connects to the broader literature on urban violence. Your comments are insightful, and we are committed to addressing the concerns raised to strengthen the manuscript.

We acknowledge your concern that the argument could be more robustly developed theoretically. In our revision, we revisit the theoretical framework to ensure that the key arguments are clearly articulated and better grounded in the existing literature. We provide a more thorough explanation of the mechanisms linking geographic patterns of violence to per capita outcomes and refine how we position our work within the broader theoretical context.

Additionally, we take seriously the observation that some claims in the manuscript may be overly assertive, given the limitations of the data and identification strategies. In our revision, we explicitly address these limitations, incorporating additional discussion of potential data and identification issues that could influence the interpretation of our findings. Furthermore, we temper any conclusions that may currently appear overly definitive, ensuring they are consistent with the scope of our analysis. To address these concerns, we explore additional robustness checks (related to R2.5) and sensitivity analyses, where possible, to provide more substantial empirical support for our conclusions. This includes a bootstrapping technique to detect if our results are only due to a few large cities, as well as a more explicit acknowledgement of the limitations of our approach where they cannot be resolved within the scope of the current study.

While acknowledging these limitations, we believe our findings make a significant contribution to the field by challenging the assumed universality of scaling “laws” of crime based primarily on population size to explain a superlinear rate. The existing literature on urban scaling of crime has predominantly focused on high-income locations, leading to conclusions about universal patterns that may not hold in different contexts. Our study provides evidence that, in African cities, geographic isolation plays a crucial role in violence patterns, a factor that has been largely overlooked

in traditional scaling analyses. This suggests that the relationship between city size and violence is more complex and context-dependent than previously thought, particularly in developing regions, where isolation can significantly influence urban dynamics. These insights, even if correlational, open new avenues for understanding urban violence beyond the conventional population-centric framework.

We are grateful for the opportunity to revise and improve the manuscript based on your helpful comments. Your feedback is instrumental in refining the theoretical framework, ensuring more measured claims, and addressing potential data and methodological challenges.

Reviewer (R2.2): *A first point is that the concept of violence is used without clear conceptual definition. In the introductory section, “violence” is extremely broad. Do you include, eg domestic violence? The references in this section are mainly about political and gang violence.*

Thank you for raising this important point about the conceptual definition of violence. We acknowledge that the introductory section could be more precise in defining the specific type of violence analysed in our study. In our work, we adopt a definition of violence that is focused on politically motivated violence against civilians. We have chosen this definition because it aligns with the scope of the ACLED data and the questions we seek to address in the manuscript. While other forms of violence, such as domestic or interpersonal violence, are important and warrant separate studies, they fall outside the framework of this analysis and outside of what the data can do. Comparing other types of violence is challenging, particularly for African countries, where data is scarce and where international comparisons would be almost impossible.

We have revised the introduction to explicitly state this definition and ensure that our focus on politically motivated violence is evident from the outset. Additionally, we have added our analysis on protests and demonstrations (which are not considered part of the violent events), related to the question R2.4 below.

Thank you again for highlighting this issue. We believe this clarification improves the manuscript’s framing and coherence.

Reviewer (R2.3): *Situating the argument, you say on p. 1 that “Although rural and sparsely-populated areas might be perceived as conducive to insurgencies due to limited state reach, they rarely offer a rich set of targets for violent activities. Violence mainly manifests as an urban phenomenon.” Firstly, is it really proven that “violence” mainly takes place in cities)? A significant literature has questioned the ability to draw such conclusions, given the systematic reporting biases that make urban violence more likely to get reported (see e.g. Dietrich & Eck 2020; Von Borzyskowski & Wahman 2021; Weidmann 2015). Secondly, rural areas are not “empty” of possible targets; they contain populations, security infrastructure, symbolic cultural sites, etc. Clarify what you mean here.*

Thank you for this thoughtful critique. You raise an important point about the perception of violence as primarily an urban phenomenon and the potential biases in reporting that might influence this conclusion. In fact, our results point in that direction as well. Violence was assumed to be mostly urban and corresponding to

big cities, but here we are showing that it is more frequent in small and isolated areas, which are not part of the core urban area.

While it is often speculated that violence predominantly occurs in urban areas, particularly in large cities, this is not consistent with what we observe in our analysis. Cities may indeed host a concentration of high-profile incidents, but our results suggest that violence can also be distributed across smaller urban agglomerations and rural areas, mainly where politically motivated dynamics create opportunities or incentives for violent actions. We have revised this section to explicitly discuss these nuances and provide a more balanced view that acknowledges the complexities of violence distribution across urban and rural areas. Further, we have expanded our literature as suggested Dietrich and Eck (2020); Von Borzyskowski and Wahman (2021); Weidmann (2015).

Furthermore, we agree that rural areas are not “empty” of potential targets. We have clarified that our point is not to suggest a lack of targets in rural settings but rather to highlight the differences in the density and type of targets between urban and rural areas. Rural areas, for example, may host populations, security installations, or cultural sites that are indeed significant targets, but these may not always provide the same level of visibility or strategic value as urban settings.

We agree that systematic reporting biases, as highlighted in the literature, need to be carefully considered when interpreting patterns of violence.

Thank you again for pointing out these issues. Addressing them will help refine our argument and ensure it is more precise and grounded in the existing literature.

Reviewer (R2.4): *The argument/relationship under study is not fleshed out theoretically. To understand why isolated cities should be at higher risk of violence, the argument about targeting necessitates clarification on exactly what form(s) of violence you include. Is it specifically violence by organized actors with political motives? At the end of the paper, you clarify that “riots and protests are not considered, as they are not typically caused by politically-motivated violent groups” (aren’t they?) – this information should come much earlier and be explicitly theorized. It seems to imply that your argument is much more limited than the front end of the manuscript suggests. Is your argument specifically about violence within a broader context of ongoing civil war? It also appears that testing the argument could be difficult due to possible reverse causality, whereby cities located in contested regions of a state also become more isolated.*

We appreciate your observations about the need for greater theoretical clarity and agree that some aspects of the argument could be presented more explicitly.

Regarding the forms of violence included. In our study, we focus specifically on violence by organized actors with political motives. As noted in the introduction, riots and protests are omitted because they typically involve different dynamics and are not usually the result of actions by violent groups. While riots and protests can sometimes have political motivations, they are distinct phenomena from the organized and targeted violence that we are examining. For example, protests against violence, which happened in Abuja to demand security, are peaceful expressions of the feelings of the population. To avoid any confusion, we ensure this exclusion is emphasized earlier and more explicitly in the introduction, alongside our definition of politically motivated violence. However, we add the analysis for protests and riots separately in the Supplementary Materials, and mention briefly the outcome in the manuscript.

Our argument is not situated within the context of an ongoing civil war but rather in areas where the lack of adequate state control creates conditions conducive to violence by politically motivated groups. This distinction is central to our theoretical framework, and we work to make it more explicit throughout the manuscript to avoid misinterpreting the scope of our study.

We also agree with the reviewer that reverse causality is a potential issue—cities located in contested regions may become more isolated as a consequence of violence rather than isolation directly leading to violence. We acknowledge this as an essential consideration which, unfortunately, we cannot quantify or fix. Historical data on road networks and travel patterns in Africa is minimal, and the temporal evolution of city isolation is poorly documented. Even when violent events cause changes in transportation infrastructure or travel patterns, these changes are rarely recorded systematically. In our work, the level of isolation of a city is measured either in the number of roads or the estimated number of journeys through that city, but both metrics are static and calculated for a single year. Using our data, we cannot see the process of a violent location becoming more isolated (either through the reduction of roads or journeys). We have highlighted this potential circular effect in the manuscript discussion.

***Reviewer (R2.5):** The research design is in many ways impressive, but as a reader, I want a lot more critical discussion about the data used (both on violence, and on cities), the possible biases and limitations associated with that data, and the implications for your analysis and results. ACLED draws on news data and is, as such, vulnerable to reporting biases (see the sources referenced below; in addition, Eck 2012). More specific to the argument presented here, is it possible that in larger cities, “everyday/routine” violence (e.g. criminal violence in slum settlements) goes unreported to a greater extent? When it comes to the city population data, how reliable is it? See e.g. Satterthwaite 2010 for a critical discussion.*

We appreciate the references you have suggested, and we have incorporated Eck (2012) and Satterthwaite (2010) into the revised manuscript to enrich our discussion of these issues.

Particularly, we agree that observing events and counting those inside different polygons is a problematic technique, particularly for small cities, as highlighted by Satterthwaite (2010). The year in which the delineation was constructed does not correspond to the events, so there are many issues regarding the classification. We have dropped that text from the manuscript but kept it in the Supplementary Materials to highlight the challenge of trying other techniques.

We agree that the use of ACLED data, which relies on media reports, is susceptible to reporting biases. However, we believe that these biases may operate in the opposite direction from what is suggested. For example, by comparing news articles in Mexico and assigning them to cities when they are being discussed, we detected a superlinear correlation with city size and media attention Prieto-Curiel et al. (2019). Additionally, we showed that although there might be shocks in the system, it tends to return to the same superlinear behaviour after a few weeks. Thus, at least in Mexico, large cities capture more media attention. Specifically, we argue that smaller and more remote areas are more likely to suffer from underreporting. In such areas, the lack of consistent media presence, as well as the absence of videos or images that can be shared widely, reduces the likelihood that national or international newspapers pick up incidents. In contrast, violence in larger cities

is more likely to attract media attention due to their higher visibility and the presence of better communication infrastructure. We expand on this argument in the manuscript and acknowledge the potential complexities while also discussing how this underreporting might influence our findings.

We test this hypothesis by considering the observed parameters of the model when the largest cities are dropped from the analysis. A method to detect whether results depend on only a few cities is through the study of randomly generated samples Cabrera-Arnau and Bishop (2020). With the inclusion or exclusion of dragon kings in the scaling analysis through sampling, the impact of a few cities on the estimated scaling parameters was detected. If our results are dependent on only a few cities, then the scaling coefficients should change rapidly when they are dropped from the analysis. Results show that this is not the case, so we are confident that our results are not dependent on a few large cities exclusively. We have added the subsection *Are the scaling coefficients the impact of only a few cities?* in the Supplementary Materials to expand on this topic and mentioned it shortly in the Results.

By incorporating these critical reflections and the suggested references, we aim to present a more balanced discussion of our data and its limitations, strengthening the overall robustness and transparency of the manuscript.

Reviewer (R2.6): *Your empirical focus on Africa is not motivated or discussed. Do you expect relationships between geographic isolation and violence to be different in Africa than in other regions? Are there important scope conditions for your argument? Discussing the connectivity component of the analysis, on page 5, you note that “city connectivity explains the emergence of dominant cities and their economic development and innovation patterns. Isolation is one of the main contributors to poverty and a violence generator.” So, is it mainly a grievance argument?*

Our focus on Africa is driven by both the availability and granularity of data, as well as the specific challenges of state capacity and geographic connectivity that are particularly pronounced in many African countries. While we do not claim that the relationship between geographic isolation and violence is unique to Africa, the continent provides a valuable context for studying these dynamics due to its diversity of political, economic, and geographic conditions. In the revised manuscript, we explicitly discuss why Africa is an appropriate case for this analysis. We acknowledge that future research could explore whether these patterns hold in other regions, such as cartel violence in Latin America. A similar project, detecting the location of cartels and the implications regarding geographical isolation, is of interest and is one of the next steps in our project.

We agree that the scope conditions of the argument should be clearly articulated. The argument is particularly relevant in contexts where state capacity is weak or unevenly distributed and where geographic isolation exacerbates vulnerabilities to violence by politically-motivated groups. These conditions are not exclusive to Africa but are pronounced in many parts of the continent, making it a suitable focus for this study. We emphasize these scope conditions in the revised manuscript to make the theoretical framing more precise.

Finally, the connection between isolation and violence in our analysis is not solely a grievance argument. While geographic isolation may contribute to grievances linked to poverty and underdevelopment, it also facilitates operational challenges for state actors in providing security and governance, which creates opportunities for politically motivated violence. The lack of connectivity can limit state presence, making

isolated areas more susceptible to violent actors who exploit this vacuum. In the revised text, we have expanded on this dual dynamic, clarifying that our argument incorporates both grievance and opportunity structures as pathways linking geographic isolation to violence. We appreciate your suggestions and will incorporate this feedback to strengthen the theoretical grounding and contextualization of our study. Thank you for helping us improve the clarity and depth of the manuscript.

Reviewer (R2.7): *In sum, the manuscript makes a strong causal claim, but neither theorizes a clear causal argument nor presents a convincing causal identification strategy.*

We appreciate your concern regarding the clarity of the causal argument and the robustness of the identification strategy. The manuscript outlines a correlation between geographic isolation and violence, and we acknowledge that the causal mechanisms could be many, but are impossible to test given the nature of the research. In the revised version, we theorize in the discussion the pathways through which isolation contributes to violence, emphasizing two possible reasons: grievance-based dynamics (e.g., poverty and lack of access to resources) and opportunity-based mechanisms (e.g., weak state presence in isolated areas creating opportunities for violent actors). By providing a more detailed explanation of these mechanisms, we aim to strengthen the theoretical foundation of the manuscript.

Furthermore, we explicitly discuss the limitations of the data and methods, ensuring that our claims are appropriately nuanced and consistent with the strength of the evidence. By refining both the theoretical and empirical components, we aim to present a clearer and more robust argument in the revised version.

Reviewer (R2.8): *Smaller comments. Discussing previous research on city size and violence (p. 2), you note contradictory findings, and conclude that “city size has been wrongfully assumed to capture violence heterogeneities, and large cities have also been wrongfully deemed to be more violent than small cities.” This is a strong claim; if taking the analytical approach in your paper (focusing on per capita levels), it seems some of the research points in that direction, but overall, the current state of knowledge cannot be summed up in that claim.*

We agree that the current state of knowledge on this topic is complex and cannot be conclusively summarized in the claim we made. We appreciate this opportunity to refine our argument. In the revised manuscript, we change the language to acknowledge the diversity of findings in the existing literature. While our analysis, which focuses on per capita levels of violence, supports the notion that large cities are not uniformly more violent than smaller cities, we recognize that other studies have found varying results depending on the context, type of violence, and methodological approach.

Reviewer (R2.9): *On p. 4, you note that “The 5% most violent cities in Africa had 73% of the fatalities related to the events in the continent since 2000 but is where only 15% of Africa’s urban population is currently living.” This makes me wonder if there are a few key cases that drive the results to a high extent (eg a few highly violent cities in the context of Ethiopia’s civil wars)? Show the reader an*

overview of the most affected cities, and discuss (and analyse) the impact of outlier cases.

Thank you very much for your comments. This issue also relates to the questions R1.2 and to R2.5 addressed previously. We consider that our results are not because of a few cities that have an abnormal level of violence, but rather, we are presenting a more general pattern observed across the continent. Thus, it is not a few very affected cities. In the manuscript, we describe how those with a high level of violence are more prevalent in the small and isolated cities, rather than the central big metropolitan areas.

Additionally, we detect if the results depend on only a few cities by dropping some of them and recompute all values of the parameters. Some cities with large population sizes (known as dragon-kings), could affect the results when estimating the scaling law parameters Arcaute et al. (2015). A method devised to detect whether results depend on only a few large cities is through the analysis of randomly generated samples Cabrera-Arnau and Bishop (2020). To address the concern about outliers, our strategy is to randomly drop a set of cities from our analysis and re-estimate the results. By systematically removing different subsets of cities, we can assess whether the main conclusions are sensitive to the inclusion of highly violent cities. This method allows us to determine if a few outlier cities primarily drive the findings or if the observed patterns hold more broadly across different urban contexts. We have added the subsection *Are the scaling coefficients the impact of only a few cities?* in the Supplementary Materials to expand on this topic and mentioned it shortly in the Results. For this analysis, we have also dropped the largest cities from the analysis and kept 45% of the urban population in the extreme, showing similar sublinear results.

This approach will provide a more rigorous test of the influence of outliers and ensure that a small number of extreme cases does not unduly shape our results. Thank you for your consideration, and we believe this addition will enhance the manuscript.

Reviewer (R2.10): *I like the careful operationalisation of connectivity and believe it is a key contribution of the paper. However, is it not likely to capture remoteness in roughly the same way as previous work relying on travel time to capital, proximity to borders (being on the “edge of the state”) etc? More discussion and reference to this literature (e.g. Fearon & Laitin and Müller-Crepon et al that you cite earlier; perhaps also Buhaug 2010), explaining how your operationalisation differs, would be useful.*

Thank you for your positive feedback on the operationalization of connectivity and for highlighting its importance as a key contribution. We agree that traditional measures of remoteness, such as travel time to the capital or proximity to borders, might capture some of the same aspects of geographic isolation. However, our approach focuses specifically on connectivity within urban systems rather than remoteness in the traditional sense. By emphasizing the connectivity of cities to each other and to economic, political, and communication networks, we aim to capture a more dynamic aspect of geographic isolation—one that goes beyond mere distance and considers how cities are integrated into broader social, economic, and infrastructural networks.

In the revised manuscript, we included a more detailed discussion of how our operationalization differs from traditional measures. We also reference the literature

you mentioned (Fearon and Laitin (2003), Müller-Crepon et al. (2021), and Buhaug (2010)) and clarify that while their measures focus on static geographic isolation or proximity to state boundaries, our measure focuses on the connectivity between cities and the broader networks they belong to. Additionally, we comment that the travel time to the capital has been proposed, Müller-Crepon et al. (2021). However, that method ignores cross-border interactions and urban corridors (such as the corridor Lagos-Accra) and changes substantially if we consider the political capital of a country or the biggest city (like Yamoussoukro, the capital of Ivory Coast or Abidjan, its largest city with 30 times more population, or like Abuja and Lagos in Nigeria).

This distinction allows us to more precisely capture the effect of isolation from broader urban and economic networks, which can contribute to violence through both material deprivation and weak state control.

REFERENCES

- Alves, L. G., Ribeiro, H. V., Lenzi, E. K., and Mendes, R. S. (2013). Distance to the scaling law: a useful approach for unveiling relationships between crime and urban metrics. *Plos One*, 8(8):e69580.
- Arcaute, E., Hatna, E., Ferguson, P., Youn, H., Johansson, A., and Batty, M. (2015). Constructing cities, deconstructing scaling laws. *Journal of the Royal Society Interface*, 12(102):20140745.
- Bettencourt, L. M., Lobo, J., Helbing, D., Kühnert, C., and West, G. B. (2007). Growth, innovation, scaling, and the pace of life in cities. *Proceedings of the National Academy of Sciences*, 104(17):7301–7306.
- Bilal, U., de Castro, C. P., Alfaro, T., Barrientos-Gutierrez, T., Barreto, M. L., Leveau, C. M., Martinez-Folgar, K., Miranda, J. J., Montes, F., Mullachery, P., et al. (2021). Scaling of mortality in 742 metropolitan areas of the Americas. *Science Advances*, 7(50):eabl6325.
- Buhaug, H. (2010). Dude, where’s my conflict? LSG, relative strength, and the location of civil war. *Conflict Management and Peace Science*, 27(2):107–128.
- Cabrera-Arnau, C. and Bishop, S. R. (2020). The effect of dragon-kings on the estimation of scaling law parameters. *Scientific Reports*, 10(1):20226.
- Dietrich, N. and Eck, K. (2020). Known unknowns: media bias in the reporting of political violence. *International Interactions*, 46(6):1043–1060.
- Eck, K. (2012). In data we trust? a comparison of UCDP GED and ACLED conflict events datasets. *Cooperation and Conflict*, 47(1):124–141.
- Fearon, J. D. and Laitin, D. D. (2003). Ethnicity, insurgency, and civil war. *American Political Science Review*, 97(1):75–90.
- Müller-Crepon, C., Hunziker, P., and Cederman, L.-E. (2021). Roads to rule, roads to rebel: relational state capacity and conflict in Africa. *Journal of Conflict Resolution*, 65(2-3):563–590.
- OECD/SWAC (2018). Africapolis (database). www.africapolis.org. Accessed: September 2019.
- Prieto-Curiel, R., Cabrera Arnau, C., Torres Pinedo, M., González Ramírez, H., and Bishop, S. R. (2019). Temporal and spatial analysis of the media spotlight. *Computers, Environment and Urban Systems*, 75:254–263.
- Satterthwaite, D. (2010). *Urban myths and the mis-use of data that underpin them*. Number 2010/28. WIDER working paper, Helsinki, Finland.
- Schläpfer, M., Bettencourt, L. M., Grauwin, S., Raschke, M., Claxton, R., Smoreda, Z., West, G. B., and Ratti, C. (2014). The scaling of human interactions with city size. *Journal of the Royal Society Interface*, 11(98):20130789.
- Von Borzyskowski, I. and Wahman, M. (2021). Systematic measurement error in election violence data: Causes and consequences. *British Journal of Political Science*, 51(1):230–252.
- Weidmann, N. B. (2015). On the accuracy of media-based conflict event data. *Journal of Conflict Resolution*, 59(6):1129–1149.

Point by point answer

REVIEWER 1

REVIEWER 2

***Reviewer (R2.1):** As I noted earlier, this is a study on an interesting and relevant topic, featuring a novel and thorough analysis of empirical patterns. In the revised manuscript, the authors have addressed many of my key concerns about causal inference and the theoretical argument. Notably, the focus on Africa is well motivated (and it is clear that there is an empirical research gap here). However, I still see a few important areas that need to be unpacked or clarified before the manuscript is suitable for publication. First, while the authors have clarified the discussion of violence in parts, it still remains unclear to the reader at the outset of the manuscript what forms of violence are in focus. In the first paragraph, the authors speak of ‘violence’ in a very broad sense (and make rather sweeping claims about how different forms of violence characterize different regions). It is not clear to the reader if claims about violence being an urban phenomenon encompass all these forms of violence. Importantly, the literature most strongly connected to this claim is the criminological literature (the references cited on page 4 in connection to the ‘universal assumption’, and most or all of the works producing scaling coefficients in SM-C are about criminal violence). Either the authors need to back up if and how this assumption has also been made with reference to political violence, or they should explicitly review literature on the urban/rural dynamics of political violence to situate their study and contribution in that literature.*

Thank you for your detailed and thoughtful feedback on the manuscript. We acknowledge your concern that the argument could be more robustly developed theoretically. We are grateful for the opportunity to revise and improve the manuscript based on your helpful comments.

We have revised the introduction to clearly define the specific forms of violence addressed in the study, distinguishing between criminal and political violence. We have clarified that “urban violence” originates primarily in the criminological violence, rather than in political violence.

To better situate our contribution, we expanded our motivation to consider the urban-rural dynamics of political violence first. A weak state in fragile urban settings enables the expansion of non-state armed groups and drives political violence, particularly in cities near conflict zones [1]. However, armed conflict has not systematically shifted toward urban areas over time. Rather, violence remains highly heterogeneous across different parts of the continent [2]. However, urban populations might experience a combination of elements which makes them vulnerable and exposed actors of armed conflict and security. For example, urban residents in Kenya had significantly lower levels of trust in the police than rural residents, reflecting the way state-society relations—and thus political tensions—are shaped by urban environments [3]. Similarly, expanding cities can both trigger and mitigate violence depending on local governance quality, institutional capacity, and the ability of cities to absorb change and manage competition [4; 5].

In the introduction, we expanded our focus on the urban dimensions of political violence while underscoring the urban-centric assumptions of our research design.

Reviewer (R2.2): *In addition to clarifying the contribution, the authors should be clear on what form(s) of political violence are in focus in the empirics. The emphasis shifts in different places, from political violence broadly (e.g. page 3) to violence against civilians specifically (e.g. in the abstract). In the methods section (p. 3), it sounds like all the categories of ACLED violence are included except protests and riots. However, when discussing the results, you focus on violence against civilians. Do you zoom in on this category as a sub-analysis, and if so, why? Motivate this upfront.*

Thank you for highlighting this issue. Indeed, we wanted to focus mostly on violence against civilians and including riots and protests in the analysis is problematic for many reasons. For example, some protests in Nigeria are precisely against Boko Haram and claiming security, or the recent demonstrations over the rising cost of living. They are not forms or expressions of political violence. However, we also think that the protests particularly help to understand the results better. The number of protests per city is close (or closer than any other type of event) to being linear with respect to size. So we clarified in the manuscript that they are dropped from the analysis, but a separate indicator and more detailed analysis are in the Supplementary Material (I - Protests and riots).

Reviewer (R2.3): *I also believe more critical discussion of the data is needed. The authors have included a discussion of the possible news biases, which is good. However, another key dimension is coding precision, which only comes up as part of the appendix. The authors have included Eck's (2012) critical comparison of ACLED and UCDP but only cursively engage with it (she reports quite large quality issues with ACLED, notably regarding subnational dynamics of violence). These issues may have improved since, but at the least, the authors should motivate why they prefer ACLED over other georeferenced violence datasets, and critically discuss possible limitations. Importantly, how many of the events are precisely coded (in terms of geoprecision) and could this affect patterns across central and more marginalized areas? In the codebook, ACLED notes that "If the source material indicates that activity took place in a small part of a region, and mentions a general area, the event is coded to a town with geo-referenced coordinates to represent that area, and the 'Geo-precision' code 2 is recorded" (p. 36). How frequent is this type of precision code, and could it affect the results (if e.g. many events in marginalised regions are coded to the nearest town)?*

We have expanded our discussion of data limitations to more fully address coding precision both in the manuscript and in the SM A - The location of events and the delineation of cities. We now engage more directly with Eck's (2012) critique, acknowledging the historical quality concerns with ACLED and noting improvements in more recent data releases.

Understanding the dynamics among violent groups is fundamental for assessing their influence on violence and governance. Three primary sources of data could be used to detect the dynamics of urban violence and its links with isolation: the Uppsala Conflict Data Program (UCDP) [6; 7], the Global Terrorism Database (GTD) [8] and the Armed Conflict Location and Event Dataset (ACLED) [9]. The three datasets are conflict event datasets based on media reports [10]. However, there are some differences between them. First, the UCDP also consults local reports and data from NGOs and international organisations like the UN, as well as local events. For example, between 2013 and 2016, approximately 20% of the events in the UCDP were reported from non-media sources, thus improving potential media

biases [6]. It provides a valuable dataset, but on the downside, UCDP has reports until 2023 and has a limited number of events compared to ACLED. Similarly, GTD does not capture protests and riots (also found here to be sublinear in terms of the number of events and casualties). Further, it has been observed that GTD captures roughly half of the terrorist events from ACLED but captures similar patterns and trends [11].

On the other hand, ACLED has updated events until May 9, 2025 (a delay of less than 10 days), and provides a much larger volume of events. Consider, for example, all events in Africa reported in 2023. ACLED has over 43,000 events, whilst UCDP has approximately 4,200 events. Thus, ACLED provides more volume and up-to-date information.

We justify our choice of ACLED over alternatives based on its broader geographic and temporal coverage, and its finer-grained actor and event distinctions, which are crucial for our analysis.

We have also added a breakdown of geo-precision codes in the data, highlighting the share of events with code 2 and 4, and we discuss how this may affect spatial patterns, particularly in some regions. The share of violence against civilians events with $geo_precision = 1$ has declined over time, from levels reaching up to 80% in the early 2000s to consistently below 60% since 2020. However, this steady decline likely does not reflect that events are more dispersed or more challenging to capture, but rather, that ACLED has improved its efforts to geocode events and report the precision more accurately. While geo-precision for protests and riots remains very high, above 95% of events are coded at the most precise level, the share is considerably lower for battles, with only around 60% of events recorded with exact location information. Additionally, there are significant geographic differences: in Nigeria, only 55% of violence against civilians events are coded with $geo_precision = 1$, whereas in Zimbabwe, the figure exceeds 95%, reflecting variation in reporting capacity and data quality across contexts.

Also, we precisely challenge the use of the precise coordinates of ACLED. As shown in the Supplementary Material Figure 1, we highlight how in Nouakchott, there are over one thousand events that share the same coordinates and they mostly have a geoprecision of 1. Thus, we know that these events happened in Nouakchott, but not exactly where. Because of the possible misalignment of the spatial polygons of Africapolis and the ACLED events, we constructed a different methodology.

We have added these motivations and limitations to the manuscript. Thank you for helping us improve our study and the manuscript.

REFERENCES

- [1] Antônio Sampaio. Urban drivers of political violence: Declining state authority and armed groups in Mogadishu, Nairobi, Kabul and Karachi. *International Institute for Strategic Studies*, 2020.
- [2] Emma Elfversson and Kristine Höglund. Are armed conflicts becoming more urban? *Cities*, 119:103356, 2021.
- [3] Emma Elfversson, Thao-Nguyen Ha, and Kristine Höglund. The urban-rural divide in police trust: insights from Kenya. *Policing and Society*, 34(3):166–182, 2024.

- [4] Emma Elfversson, Kristine Höglund, Angela Muvumba Sellström, and Camille Pellerin. Contesting the growing city? forms of urban growth and consequences for communal violence. *Political Geography*, 100:102810, 2023.
- [5] Emma Elfversson and Kristine Höglund. Urban growth, resilience, and violence. *Current Opinion in Environmental Sustainability*, 64:101356, 2023.
- [6] Ralph Sundberg and Erik Melander. Introducing theUCDP georeferenced event dataset. *Journal of Peace Research*, 50(4):523–532, 2013.
- [7] Shawn Davies, Garoun Engström, Therese Pettersson, and Magnus Öberg. Organized violence 1989–2023, and the prevalence of organized crime groups. *Journal of Peace Research*, 61(4):673–693, 2024.
- [8] GTD Global Terrorism Database 1970 2020. National Consortium for the Study of Terrorism and Responses to Terrorism, 2022.
- [9] Clionadh Raleigh, Andrew Linke, Håvard Hegre, and Joakim Karlsen. Introducing ACLED: an Armed Conflict Location and Event Dataset: special data feature. *Journal of Peace Research*, 47(5):651–660, 2010.
- [10] Kristine Eck. In data we trust? a comparison of UCDP GED and ACLED conflict events datasets. *Cooperation and Conflict*, 47(1):124–141, 2012.
- [11] Rafael Prieto-Curiel, Olivier J Walther, and Ewan Davies. Detecting trends and shocks in terrorist activities. *PloS One*, 12(6):e0179057, 2023.